# Cell-to-Cell Interactions during Early *Drosophila* Oogenesis: An Ultrastructural Analysis

**DOI:** 10.3390/cells11172658

**Published:** 2022-08-26

**Authors:** Maria Giovanna Riparbelli, Veronica Persico, Giuliano Callaini

**Affiliations:** Department of Life Sciences, University of Siena, 53100 Siena, Italy

**Keywords:** *Drosophila* oogenesis, egg chambers, follicle cells, oocyte, asymmetry

## Abstract

*Drosophila* oogenesis requires the subsequent growth of distinct egg chambers each containing a group of sixteen germline cells surrounded by a simple epithelium of follicle cells. The oocyte occupies a posterior position within the germ cells, thus giving a distinct asymmetry to the egg chamber. Although this disposition is critical for the formation of the anterior–posterior axis of the embryo, the interplay between somatic and germ cells during the early stages of oogenesis remains an open question. We uncover by stage 2, when the egg chambers leaved the germarium, some unique spatial interactions between the posterior follicle cells and the oocyte. These interactions are restricted to the surface of the oocyte over the centriole cluster that formed during early oogenesis. Moreover, the posterior follicle cells in front of the oocyte display a convoluted apical membrane with extensive contacts, whereas the other follicle cells have a flat apical surface without obvious surface protrusions. In addition, the germ cells located at the posterior end of the egg chamber have very elongated protrusions that come into contact with each other or with facing follicle cells. These observations point to distinct polarization events during early oogenesis supporting previous molecular data of an inherent asymmetry between the anterior and the posterior regions of the egg chambers.

## 1. Introduction

The *Drosophila* ovary consists of two symmetric groups of 16–18 ovarioles. Each ovariole is composed of a germarium followed by a chain of developing egg chambers that contain 15 nurse cells and a developing oocyte. The egg chambers mature moving from the anterior to the distal region of the ovariole through fourteen developmental stages defined by their relative position within the ovariole, their distinct morphology and the oocyte differentiation [1]. The oogenesis begins in the anterior region of the ovariole, the germarium, that holds somatic and germline stem cells and has a stereotypical organization defined by the stage of germ cell development with four morphologically distinct regions: regions 1, 2a, 2b, and 3, or stage 1. Two to three germline stem cells (GSCs) divide asymmetrically at the tip of the germarium to self-renew and give origin to daughter cystoblasts. The cystoblasts undergo four incomplete mitotic divisions to form in region 2a distinct cysts of 16 germ cells connected by cytoplasmic bridges, the so-called ring canals [2]. The germ cells look the same, but the four incomplete mitotic divisions lead to cells linked by a different number of ring canals. In particular, the two older sister cells that originate by the first division of the cystoblast, the pro-oocytes, have four ring canals. The pro-oocyte that inherits from the cystoblast an unusually long mother centriole [3] and the main part of the spectrosome [4] will be the differentiating oocyte, whereas the sister cell will turn into the 15-nurse cell cluster that provides the cytoplasmic components need for oocyte growth and maturation. 

After moving through region 2b of the germarium, the germ cell cysts become surrounded by a thin layer of somatic cells, the follicle cells, that take origin from a few stem cells that reside at the anterior border of the region 2a of the germarium [5,6,7,8,9,10,11,12]. The follicle stem cells divide asymmetrically to self-renew and give origin to daughters that proliferate and separate the individual germline cysts to form a uniform cuboidal epithelium in region 3, or stage 1 [13,14]. 

Starting from stage 2, the egg chambers leave the germarium and enter the more posterior region of the ovariole, the vitellarium, where the oocytes growth and complete meiosis at stage 14. The follicle cells gradually differentiate into three distinct types: polar cells that reside at the anterior and posterior ends of each egg chamber, stalk cells that connect adjacent egg chambers and main-body follicle cells that form a continuous epithelium around the germline cysts [15]. 

Although a complex sequence of signaling events and close reciprocal interactions between somatic and germline cells occur during the early *Drosophila* oogenesis [16,17,18,19,20,21,22,23,24,25] open questions remain on the structural interplay between the follicle cells and the oocyte. We examined here the morphology of the cell contacts during egg chamber formation and early stages of vitellogenesis to add insights to the general understanding of the reciprocal interaction between somatic and germ cells in *Drosophila* oogenesis. 

## 2. Materials and Methods 

### 2.1. Fly Stocks

The *Drosophila melanogaster* Oregon-R stock was maintained on standard agar-cornmeal medium in a 12/12 light/dark cycle at 24 °C.

### 2.2. Transmission Electron Microscopy

Ovaries from 4–5-day old females were dissected in phosphate-buffered saline (PBS), and fixed in 2.5% glutaraldehyde in PBS overnight at 4 °C. Samples were post-fixed in 1% osmium tetroxide in PBS for 1 h at 4 °C. The material was then dehydrated through a graded series of ethanol, infiltrated with a mixture of Epon–Araldite resin. Polymerization was performed at 60 °C for 48 h. Silver-grey sections (70 nm thick) were cut with a Reichert ultramicrotome, equipped with a diamond knife. The sections were collected with formvar-coated copper slot grids and stained with 2% aqueous uranyl acetate for 20 min in the dark and then with lead citrate for 2 min. TEM preparations were observed with a Tecnai G2 Spirit EM (FEI, Eindhoven, The Netherlands) equipped with an Osis Morada CCD camera (Olympus, Tokyo, Japan).

## 3. Results

The asymmetric division of the female GSCs in the anterior region of the ovariole gives origin to a new self-renewing stem cell and a daughter cystoblast. Small contact areas (0.10–0.13 μm length; *n* = 12) have been observed at the interface between the cystoblast and the surrounding somatic escort cells. These contacts are characterized by a weak accumulation of dense material on the inner face of the opposing cell membranes (Figure 1A). 

The cystoblast undergoes in region 2a four rounds of mitotic divisions and originates a germline cyst of 16 interconnected cells, the cystocytes. Small surface contacts (0.14–0.19 μm length; *n* = 17) were present between the cystocytes and the escort cells (Figure 1B) and between the adjacent membranes of the strictly packed laminar extensions of the escort cells that surround the cystocytes (Figure 1B). Although the morphology of these contacts is reminiscent of the typical zonulae adherentes usually found between the epithelial cells, we were unable to identify distinct microfilaments associated with the dense material underlying the opposite plasma membranes. Therefore, we regard these short contacts as basic or incipient adherens junctions (AJs) rather than true zonulae adherentes.

Broad AJs (0.35–048 μm length; n = 22) were found among the neighboring germ cells of the same cyst in region 2b (Figure 1C). In addition to the dense material on the inner face of the opposing membranes, we often observed distinct thin bridges crossing the extracellular space of the AJs (Figure 1C). Distinct AJs were also found in the proximity of the ring canals when the membranes of the adjacent cells folded and approached each other (Figure 1D). A thin epithelial sheath of somatic cells, the follicle cells, surrounded the germ cells in region 2b to complete the egg chamber. Small AJs (0.15–0.18 μm length; *n* = 25) were also found between the germ cells and the adjacent follicle cells (Figure 1D).

The oocyte, recognized by the synaptonemal complexes and a distinct cluster of centrioles within its posterior cytoplasm, contacted the apical region of the follicle cells at the posterior end of the region 3 egg chambers, or stage 1 (Figure 2A, inset). Sections through the interface between the oocyte and the posterior follicle cells at the level of the centriole cluster showed several short AJs (0.08–0.15 μm length; *n* = 26) intermingled with small narrow spaces in which the membranes of the facing cells were slightly separated (Figure 2A,A’). The spaces among the facing cells increased moving up or down with respect to the centriole cluster. The membrane of the oocyte raised here in broad and short protrusions that contacted the apical region of the posterior follicle cells with small points in which a dense material was only associated with the oocyte side of the plasma membrane (Figure 2B). The apical membrane of the follicle cells in front of the oocyte had broad and irregular folds that intermingled among them and showed extended AJs (Figure 2A’). By contrast, the anterior and the main-body follicle cells had flattened apical membranes without intertwined extensions (not shown). Surprisingly, within each cluster, we find some centriole pairs in which a procentriole was orthogonal to the basis of a mother centriole (Figure 2A’).

By stage 2, when the egg chambers leaved the germarium, the contact area between the oocyte and the posterior follicle cells had mainly restricted to the oocyte surface above the centriole cluster (Figure 2C), where AJs of different extensions were seen (Figure 2C’). Large spaces between the facing cells were evident in sections taken up or down the level of the centriole cluster. These spaces were filled by elongated projections of the follicle cells that interdigitate with short protrusions of the oocyte membrane (Figure 2D). The membrane projections of both the cell types were reduced in size and number near the marginal posterior region of the oocyte (Figure 2C). The anterior follicle cells in region 3 and stage 2 egg chambers had flattened apical surfaces without intertwined extensions (not shown). Empty spaces were present between the anterior follicle cells and the facing nurse cells (Figure 2C, inset) and only occasionally the cell membranes formed short projections that came into contact with each other. Likewise, the main body follicle cells had wavy apical surfaces and were separated by the facing nurse cells by distinct spaces occasionally interrupted by short membrane contacts. Small AJs were also found at the interface between the nurse and follicle cells (not shown).

High magnification of the apical region of the follicle cells in front of the centriole cluster highlighted some microtubules running within the membrane projections and distinct microtubule bundles ending within the dense material lining the AJs (Figure 2E). These structures were not visible in the anterior follicle cells perhaps due to the reduced size of their membrane protrusions.

Cross sections near the edges of the stage 2 egg chambers revealed large spaces among the peripheral nurse cells (Figure 2F, inset). Remarkably, the spaces at the posterior margin of the egg chamber were filled by thin elongated protrusions (2.9–3.6 μm; *n* = 32), that emerged from the germ cells and contacted each other laterally (Figure 2F). In contrast, the follicle cells at the posterior edge of the egg chamber had shorter projections that often became in contact with the long projections of the germ cells (Figure 2F’). The germ cells of the anterior region of the egg chamber did not show such elongated projections but only had a wavy surface.

Midbody remnants associated with single follicular cells have been often found, pointing to the asymmetric inheritance of this structure, (Figure 2G). Moreover, distinct midbodies crossed by longitudinal microtubules were observed between adjacent follicle cells, suggesting that the abscission at the end of telophase is delayed in these cells that maintained long time their connection (Figure 2G’). This could explain the finding of tripartite midbodies that point to subsequent mitoses without the proper separation of the sister cells (Figure 2G’’).

During stages 3–4 (Figure 3A, inset), the oocyte membrane above the centriole cluster and the membrane of the facing follicle cells formed broad protrusions that interdigitate among them (Figure 3A). Large AJs (0.31–0.49 μm length; *n* = 21) were found between the apical region of the posterior follicle cells (Figure 3A’,B) and where the protrusions of the oocyte contact the follicle cell membrane (Figure 3A’,B). Thin elongated projections of the follicle cells (1.98–2.05 μm length; *n* = 11) often entered deeply into the oocyte cytoplasm (Figure 3A’,B). Such membrane specializations were not found in the anterior region of the egg chamber (Figure 3A, inset).

During stages 5–6 (Figure 3C, inset), the contact area between the oocyte and the follicle cells was still restricted to the region above the centriole cluster, whereas the interface between the marginal posterior region of the oocyte and the facing follicle cells had large spaces (Figure 3C). Remarkably, the apical region of the follicle cells expanded into a large infolding of the oocyte surface in correspondence with the centriole cluster (Figure 3C). Here, the apical membrane of the follicle cells folded and extended in irregularly shaped protrusions that interdigitated with short projections of the oocyte or contacted the oocyte surface (Figure 3C’). Only the follicle cells facing the centriole cluster showed large projections, whereas the membrane of the follicle cells in front of the posterior margin of the oocyte showed no such apical folds, but only had short projections that often contacted the oocyte membrane (Figure 3C). Extended AJs were observed between the convoluted apical region of adjacent follicle cells and between the oocyte and the follicle cell membranes (Figure 3D). The AJs between the oocyte and the follicle cells had often an asymmetric organization. We find, indeed, some contact areas in which the dense material was only seen on the cytoplasmic face of the oocyte, whereas usually the AJs had the dense material on the facing membranes of both the adjacent cells (Figure 3D). The apical membrane of the follicle cells that surrounded the nurse cells in the mid and anterior regions of the egg chambers slightly bulged out, whereas the facing nurse cells showed feeble protrusions (Figure 3C, inset). AJs were mainly found at the apical interface of the neighboring follicle cells (not shown).

The size and number of the AJs at the interface between the oocyte and the follicle cells progressively decreased as the egg chambers grew during the following stages and the extracellular space among the facing cells concurrently increased. During stage 7, both the membranes of the posterior region of the oocyte and the apical region of the facing follicle cells rise in long, thin projections that interdigitate among them (Figure 4A). AJs were found between the apical membranes of the follicle cells (Figure 4A) but were no longer visible at the interface between the oocyte and the follicle cells. Close contacts between large protrusions of the oocyte and the follicle cell membrane were occasionally found (Figure 4B). Such contacts, characterized by thin bridges in the intercellular space, look like typical septate junctions usually found between adjacent epithelial cells. High magnification of the oocyte membrane facing the posterior follicle cells uncovered an intense endocytic membrane trafficking. Invaginating vesicles at the basis of the membrane projections and distinct vesicles of different sizes in the peripheral cytoplasm were indeed found (Figure 4C). Only the follicle cell membranes facing the posterior region of the oocyte formed thin projections. By contrast, the surface of the follicle cells surrounding the nurse cells only showed short protrusions. Likewise, the nurse cell membranes facing the follicle cells did not show distinct surface projections.

During stage 8 the remaining centrioles had migrated to the dorsal anterior region of the oocyte in front of the nurse cells (Figure 4D. The membranes of the nurse cells and the oocyte elongate in thin projections that interdigitated through the narrow space among the facing cells (Figure 4D,E). The membrane of the oocyte showed several invaginating vesicles, suggesting endocytic trafficking between the oocyte and the adjacent nurse cells (Figure 4F). Remarkably, the membrane of the nurse cells adjacent to the oocyte raised in thin projections, whereas the nurse cells that were in contact among them and with the surrounding follicle cells had a smooth surface without invaginating membrane vesicles.

## 4. Discussion

The proper positioning of the GSCs in the anterior region of the ovariole is ensured by the adhesion with a specialized kind of supporting cells, the cap cells. In particular, the GSCs are anchored to the cap cells at the apical end of the germarium by adherens [26] and gap [27] junctions. Gap junctions have been also observed in egg chambers from stages 2 to 10 [27,28]. In addition, incipient septate junctions have been described from stage 6 egg chambers [29,30]. However, gap and septate junctions, do not seem to be relevant for germ cell maintenance and dynamics in regions 2a–2b of the germarium. In region 2a, we found small contacts between the laminar extensions of the escort cells and the cystoblast and between the escort cells and the peripheral germ cell of the forming cysts. Small contacts are also seen between the peripheral germ cells of the cysts and the surrounding follicle cells in region 2b. This suggests that the positioning of the germ cells within the germarium does not need strong anchoring to the supporting cells, but only transient contacts. By contrast, extensive contacts have been observed between the germ cells inside the cysts and between the pro-oocytes. The peripheral germ cells that have small contacts can undergo spatial rearrangement and could facilitate the shaping of the cysts from round clusters in region 2a to lens-shaped clusters in region 2b. 

The short projections of the oocyte membrane that contact the follicle cells in region 3 resemble the punctum adherens found during the development of various organisms [31]. However, the punctum adherens is characterized by a dense material on both the membranes of the adjacent cells, whereas this material is present only on the oocyte side of the plasma membrane. This suggests that the junctions at the interface between the oocyte and the follicle cells may first assemble at the oocyte surface, perhaps in answer to polarized cues from the posterior follicle cells. Such a condition in which the dense material of the junction is asymmetrically distributed between the facing membranes of the opposing cells has been observed in the male stem cell niche of *Drosophila* where the dense material accumulates preferentially at the membrane of the germline stem cells rather than to the membrane of the somatic hub cells [32]. 

The contact area where the dense material is only found at one side of the opposing cells could represent an early step of the assembly of the AJs, suggesting that the completion of these heterotypic junctions requires different times within the facing cells. The assembly of the AJs presumably starts at the membrane of the germ cells upon contact with the somatic cells. Then, the thin bridges that form within the interface between the opposing cells may induce the completion of the junction at the somatic cell side. Perhaps, the junctions form firstly at the membrane side of the germ cells in answer to polarized cues from facing somatic cells.

EM analysis showed a large number of wide AJs between the convoluted apical membrane of the posterior follicle cells. This suggests that the DE-cadherins, observed at the interface between the oocyte and the posterior follicle cells [33,34], are mainly enriched to the follicle cell side, rather than to the oocyte counterpart. Moreover, the apical membrane of the lateral and anterior follicle cells is mostly flat, and the AJs are only found in the apical region of the adjacent cells. This could explain why the concentration of DE-cadherin as seen by immunofluorescence analysis is highest on the posterior follicle cells [33]. The cadherin-mediated adhesion between the follicle cells and the oocyte in the germ cell cyst in region 3 could represent the first polarization event of the oocyte to the posterior end of the egg chamber [33,35,36]. The AJs between the oocyte and the follicle cells are maintained until stages 7–8 when the oocyte and the follicle cells are closely apposed. Accordingly, the concentration of E-cadherin at the interface between the oocyte and the posterior follicle cells is maintained until stage 7 of the oogenesis [33]. 

The projections of the posterior follicle cells are mainly restricted during stages 2–5 to the membrane facing the region of the oocyte where the centriole cluster is found. This area where the membrane protrusions are most evident could be functionally related to the transfer of polarized signals from the facing cells. The elongated protrusions emerging from the follicle cells that often enter deeply into the oocyte cytoplasm could be directly involved in these signaling events. Signal delivery by membrane protrusions has been already described in *Drosophila* germaria. It has been shown, indeed, that the cap cells utilize actin-rich membrane extensions, the cytonemes, that may deeply insert in the cytoplasm of target cells to deliver Hedgehog signal [37]. Filopodia extensions from cap cells to GSCs have been also implicated in the Bone morphogenetic protein signaling pathway [38]. Moreover, the escort cells extend long protrusions to generate a dynamic compartment for GSC differentiation [39]. 

Why the most part of the protrusions at the follicle cell/oocyte interface emerges in correspondence with the centriole cluster? One trivial explanation could be that the centriole cluster gives an intrinsic asymmetry by recruiting the centrosomal material needed for the nucleation of microtubules which plus end contact with the plasma membrane of the oocyte, presumably inducing the formation of distinct surface protrusions. These protrusions come into contact with the apical membrane of the follicle cells which in turn form elongated projections probably mediated by the assembly of longitudinal microtubule bundles and the polymerization of the actin filaments. We observed, indeed, longitudinal microtubules that cross the membrane projections or end to the AJs at the interface between the follicle cells and the oocyte. An additional aspect characterizing the early *Drosophila* oogenesis is the presence of centriole pairs in which one procentriole was often found orthogonal to its mother [3]. This condition that was observed within the germarium and during early stages in the vitellarium suggests that post-mitotic centrioles can duplicate independently of the DNA replication that ceases after the fourth mitotic division of the germ cells. This is an intriguing point because canonical centriole duplication is tightly coupled with DNA replication to ensure the correct number of centrosomes during each cell cycle.

The asymmetry between the anterior and the posterior region of the egg chamber is strengthened by the different morphology of the peripheral nurse cells during stages 2–3. The nurse cells at the posterior margin of the egg chambers display very long protrusions that make contact among them or with the membrane of the adjacent follicle cells. By contrast, the nurse cells of the anterior region of the egg chamber did not display such surface protrusions. It is unclear if such structures could represent additional sites of signal delivery. 

During stages 7–8 the cadherin accumulation at the interface between the follicle cells and the oocyte has decreased dramatically [33,34] and these cells were no longer apposed to one another. Moreover, the gap junctions observed between the interdigitated microvilli of the follicle cells and the oocyte become undetectable starting from stage 9 [28]. However, small septate junctions appear during stage 8 when broad protrusions of the oocyte and follicle cell membranes become in contact. This finding is unexpected since septate junctions have been described in the *Drosophila* egg chambers, but they only assemble by stage 6 between adjacent follicle cells [30,40,41,42]. Moreover, these contacts represent an intriguing exception since the septate junction is usually found at the apical-lateral region of adjacent cells of the same tissue and are stable contacts that usually persist during the cell life. By contrast, the septate junctions we found during stage 8 appear at the interface of different cell types and early disappear. This finding raises additional questions about the actual nature and function of these junctions during oocyte development. 

Since the oocyte nucleus is transcriptionally quiescent, the growth of the oocyte mainly depends on stored mRNAs or material flowing through the ring canals of the neighboring nurse cells and by yolk proteins delivered by the follicle cells. However, the finding of endocytotic vesicles in the anterior and posterior regions of the oocyte membrane suggests that other mechanisms are required to deliver additional materials for proper oocyte development. The membrane trafficking at the posterior oocyte surface could represent, indeed, the incipient process of clathrin-mediated uptake of the first yolk proteins exocytosed by the follicle cells facing the oocyte [43,44]. Alternatively, the endocytotic vesicles might be involved in polarized secretion mechanisms between the oocyte and the facing follicle or nurse cells. However, both these possibilities need molecular data, in addition to the ultrastructural observations, to be confirmed. 

## Figures and Tables

**Figure 1 cells-11-02658-f001:**
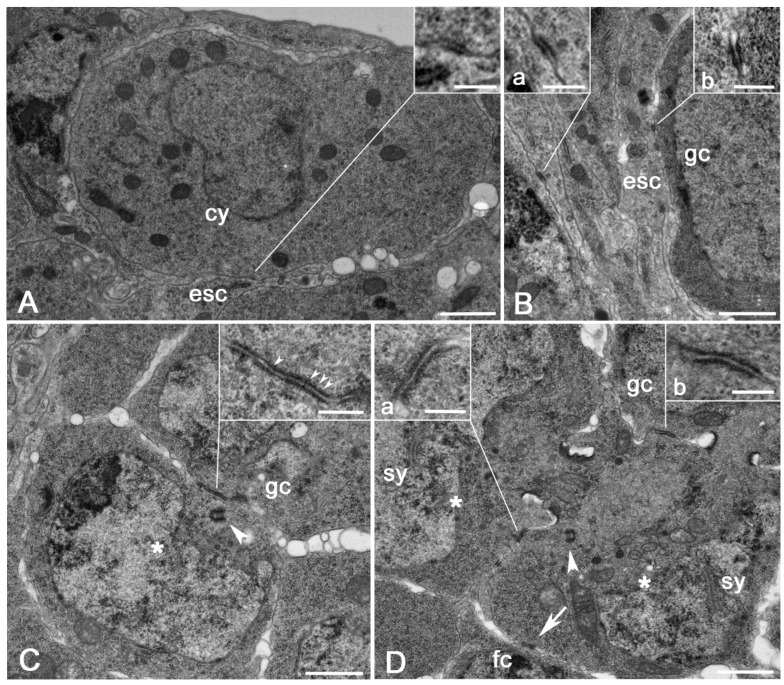
(**A**) AJs between a cystoblast and the surrounding escort cells (inset). (**B**) Region 2a: AJs between the extensions of neighboring escort cells (a, inset) and between the germ cells and the escort cells (b, inset). (**C**) Region 2b: detail of a large AJ between a pro-oocyte and a germ cell; note the thin bridges crossing the extracellular space of the junction (arrowheads). (**D**) Large AJs between two pro-oocytes (a, inset) and between a pro-oocyte and a germ cell (b, inset); the pro-oocytes are recognized by the presence of distinct synaptonemal complexes; short AJs (arrow) are also present between the pro-oocyte and the surrounding follicle cells (fc). Arrowheads point to centrioles close to the AJs; cy, cystoblast; esc, escort cells; gc, germ cells; asterisk, pro-oocyte; sy, synaptonemal complexes; fc, follicle cells. Scale bars: (**A**–**D**), 1 μm; insets, 0.2 μm.

**Figure 2 cells-11-02658-f002:**
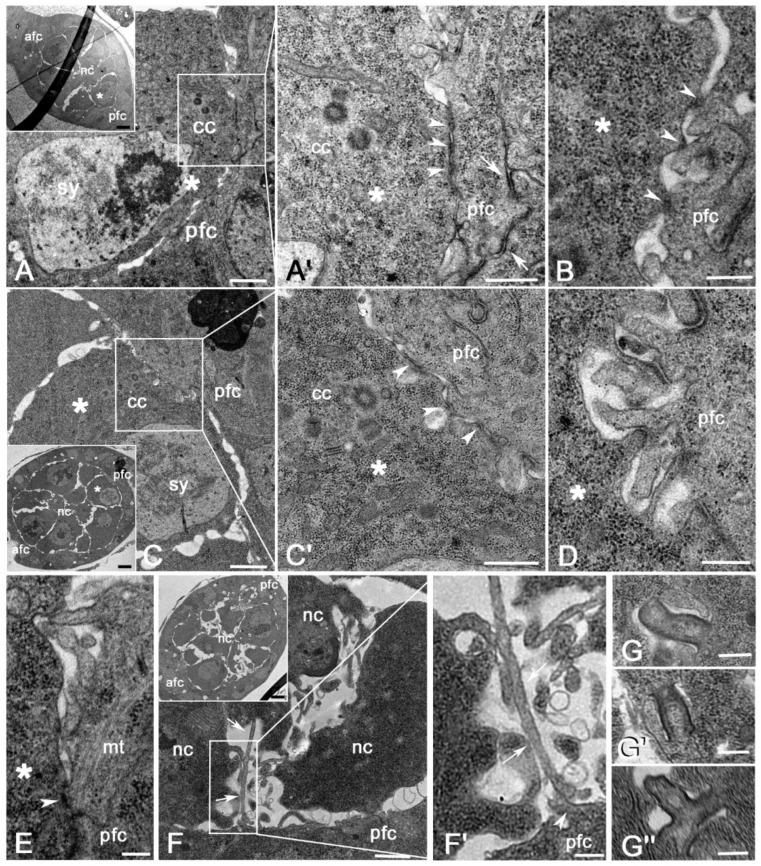
(**A**) The oocyte recognized by the synaptonemal complexes, and one cluster of centrioles contacts the follicle cells at the posterior end of the region 3 egg chamber. (**A’**) Detail of the contact region between the oocyte and the follicle cells just over the centriole cluster: several short AJs (arrowheads) are observed at the interface between the opposing cells, whereas the convoluted apical region of the follicle cells displays large AJs (arrows). (**B**) Cross section at the interface between the oocyte and the follicle cells below the centriole cluster showing short protrusions of the oocyte membrane in contact with the apical surface of the follicle cells; the protrusions of the oocyte show a dense material restricted to the oocyte side of the plasma membrane (arrowheads). (**C**) Stage 2 egg chamber: the contacts between the oocyte and the posterior follicle cells are more evident in correspondence of the centriole cluster. (**C’**) Detail of the interface between the oocyte and the follicle cells in correspondence with the centriole cluster: several short AJs are present (arrowheads). (**D**) Section below the region of the centriole cluster showing intermingled projections of the oocyte and the follicle cells. (**E**) Detail of the contact area between the oocyte and the follicle cells showing distinct microtubules ending toward the AJ (arrowhead). (**F**) Cross section and (**F’**) detail of the posterior region of a stage 2 egg chamber: the peripheral germ cells show elongated projections (arrows) that come in contact among them or with short protrusions (arrowhead) of the follicle cells. (**G**–**G’’**) Details of some midbody remnants found in the follicle cells. asterisk, pro-oocyte; sy, synaptonemal complexes; cc, cluster of centrioles; afc, pfc, anterior and posterior follicle cells; mt, microtubules nc, nurse cells. Scale bars: (**A**,**C**,**F**), 1 μm; (**B**,**D**), 0.2 μm; (**A’**,**C’**), 0.3 μm; (**E**,**F’**), 0.2 μm; (**G**–**G’’**), 0.2 μm; insets (**A**,**C**,**F**), 3 μm.

**Figure 3 cells-11-02658-f003:**
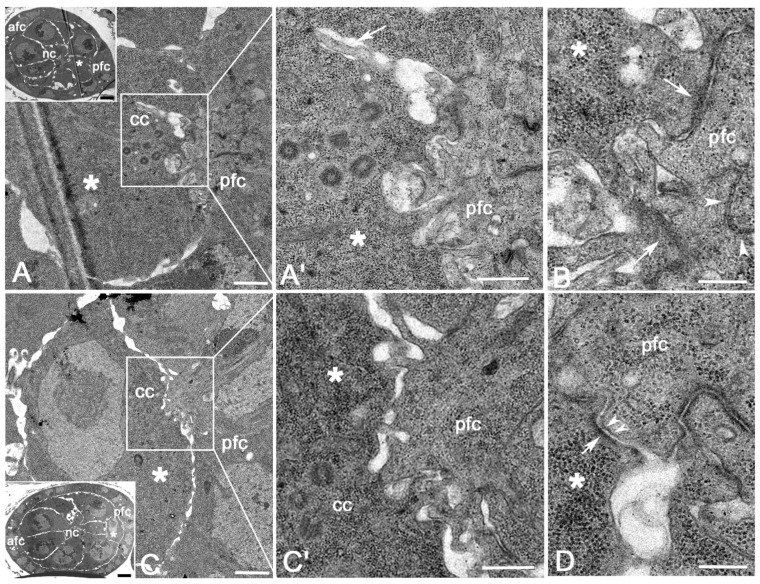
(**A**) Cross section of a stage 3 egg chamber and (**A’**) detail of the posterior region of the oocyte at the level of the centriole cluster: irregular projections of the follicle cells interdigitate with opposite projections of the oocyte membrane or enter deeply in the oocyte cytoplasm (arrow). (**B**) Magnification of the posterior region of the oocyte at the level of the centriole cluster showing large contacts between the oocyte and the follicle cell membranes (arrows) and between the convoluted apical region of adjacent follicle cells (arrowheads). (**C**) Section of the posterior region of a stage 5 egg chamber and (**C’**) detail of the interface between the follicle cells and the oocyte: the more prominent protrusions of both the follicle and the oocyte membranes are restricted to the region in front of the centriole cluster. (**D**) Magnification of an AJ between the oocyte and the facing follicle cells showing a dense material only present on the oocyte side of the plasma membrane (arrow) and distinct thin bridges (arrowheads) in the space between the membranes of the adjacent cells. asterisk, oocyte; cc, centriole cluster; nc, nurse cells; afc, pfc, anterior and posterior follicle cells. Scale bars: (**A**–**C**), 1 μm; (**A’**,**C’**), 0.3 μm; (**B**,**D**), 2 μm; inset (**A**), 3 μm, inset (**C**), 4 μm.

**Figure 4 cells-11-02658-f004:**
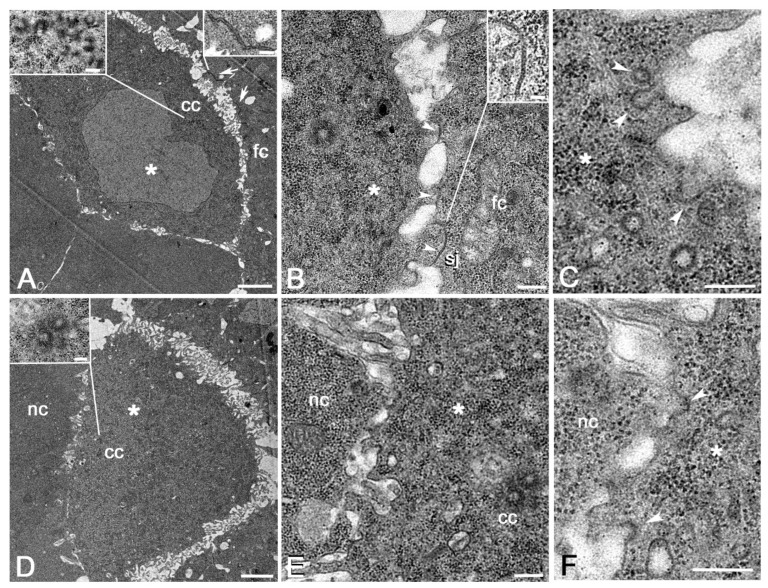
(**A**) Cross section of a stage 7 egg chamber: the space between the posterior region of the oocyte, which still maintains many centrioles, and the facing follicle cells is greatly increased; both the oocyte and the follicle cell membranes elongate in thin elongated protrusions that interdigitate among them; AJs (arrows) are found between adjacent follicle cells. (**B**) Details of some close contacts (arrowheads) between the large protrusions of the oocyte and the follicle cell membranes; a septate junction is also found between the oocyte and the follicle cells. (**C**) Magnification of the oocyte membrane facing the posterior follicle cells: several vesicles (arrowheads) arise from the plasma membrane of the oocyte. (**D**) Cross section of a stage 8 egg chamber: some centrioles are visible in the anterior dorsal region of the oocyte. (**E**) Detail of the anterior dorsal region of the oocyte at the level of the remnant centriole cluster showing thin interdigitating projections of the oocyte and the adjacent nurse cell membranes. (**F**) Magnification of the interface between the oocyte and the nurse cells showing pinocytotic vesicles emerging from the oocyte plasma membrane (arrowheads). Asterisk, oocyte; cc, centrioles; fc, follicle cells; sj, septate junction; nc, nurse cells. Scale bars: (**A**,**D**), 2 μm; (**B**,**E**), 0.2 μm; (**C**,**F**), 0.2 μm; insets, 0.2 μm.

## Data Availability

Not applicable.

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
