# Peer review of "Cell-to-Cell Interactions during Early Drosophila Oogenesis: An Ultrastructural Analysis"

_cells, 2022, doi:10.3390/cells11172658_

Round 1

Reviewer 1 Report

Ultrastructural studies of the formation of asymmetry during oogenesis and further during the development of embryos of organisms of various taxonomic groups are an important and invaluable contribution to the study of the problem of regulation of differentiation and formation of an organism from one initial cell. In this regard, the present study of the authors is undoubtedly very relevant.

I have minor comments on the text and questions for the authors of the article before the article can be published. I am sure that the answers to these questions will be of interest not only to me, but also to readers of the Cells journal.

Major points

Figure 1. In photographs 1C and 1D, it is worth noting that centrioles are located near the adherens junctions and to mark them in the photographs.

Figure 2. It is interesting to note and discuss that there is a procentriole on the centriole in photo Aá¾½, hence the centriolar cycle in the oocyte continues. How is it related to the cell (nuclear) cycle in this case? If there are serial sections of this region, does a synchronous duplication of all centrioles in the cluster occur?

How many centrioles are in a cluster  at different stages ?

Page 8

During stages 9-10 the remnant centrioles had migrated to the dorsal anterior region of the oocyte in front of the nurse cells (Figure 4D).

Centrioles were destroyed and their number decreased? What are «centriole remnants»?

Centrioles migrated from posterior region of the oocyte to the dorsal anterior region of the oocytes.

Can you propose explanation of the mechanism and meaning of this migration?

Minor points

2. Materials and methods

2.2. Transmission electron microscopy.

«Silver-grey sections...»

Such a description is understandable for electron microscopists, but for a wider readership, the thickness of the sections should be indicated

« Ultra thin silver-grey sections (thickness 70 nm) …»

“stained with uranyl acetate and lead citrate.”

What was time of staining and concentration of uranyl acetate?

Author Response

First at all we would like to thank the Reviewer for appreciating our work.

Figure 1. In photographs 1C and 1D, it is worth noting that centrioles are located near the adherens junctions and to mark them in the photographs.

As suggested by the Reviewer we mark in Fig. 1C and 1D the centrioles close to the AJs.

Figure 2. It is interesting to note and discuss that there is a procentriole on the centriole in photo Aá¾½, hence the centriolar cycle in the oocyte continues. How is it related to the cell (nuclear) cycle in this case? If there are serial sections of this region, does a synchronous duplication of all centrioles in the cluster occur?

As observed by the Reviewer there is in Fig.2A’ one mother and one orthogonal daughter procentriole. As suggested the centriole cycle in the Drosophila germarium continues despite the DNA replication cycle is arrested. We added a paragraph in the Result and Discussion sections to highlight this finding.

We have scored serial sections of different centriole clusters, and we see, indeed, some centrioles with procentrioles, but we did not observe a synchronous duplication among the centrioles of the same clusters. This suggests that duplication could also occur before the migration to the oocyte. Moreover, the growth of the procentrioles is often delayed, since we had observed centrioles of different length (Riparbelli et al., 2021).

How many centrioles are in a cluster at different stages ? Centrioles were destroyed and their number decreased? What are «centriole remnants»?

The number of centrioles within each cluster decrease during egg chamber development. Pimenta-Marques et al. (2016) on the basis of fluorescence intensity estimated an average of 50 centrioles during stages 7-8 and  8-10 centrioles during stages 12-13. Our EM observations revealed higher centriole numbers (about 40-50) during early stages (3-6) but we scored only 10-14 centrioles during stages 7-9. However, despite the different number of centrioles observed by Pimenta et al. and by us, the relevant finding, as the Reviewer observed, is that the number decreases with age.

The decrease in number is likely due to their disassembly/degeneration. We find, indeed, during later stages centrioles with defects in their wall, suggesting a probable disassembly (Riparbelli et al., 2021).

With “remnants” we would like to point to centrioles still visible during later stages. However, we changed with “remaining”

During stages 9-10 the remnant centrioles had migrated to the dorsal anterior region of the oocyte in front of the nurse cells (Figure 4D).

Centrioles migrated from posterior region of the oocyte to the dorsal anterior region of the oocytes.

Can you propose explanation of the mechanism and meaning of this migration?

The mechanism of centriole migration is still understood. The early migration through the ring canals that occurs within the germarium seems to be independent by intact microtubules, but it is associated with dynein that acts along microtubules. Perhaps there are colcemid-resistant microtubules along which the centrioles can move. Also, the migration from the posterior region of the oocyte towards its dorsal-anterior region is unclear. Moreover, the meaning and function of the centrioles during oogenesis is still debating, since Sas4  female flies that lack centrioles laid normal eggs.

«Silver-grey sections...» Such a description is understandable for electron microscopists, but for a wider readership, the thickness of the sections should be indicated.

We specify the thickness of the sections as suggested.

What was time of staining and concentration of uranyl acetate?

Apologies for the absence of details that it is correct to include. We change the paragraph: “ stained with 2% aqueous uranyl acetate for 20 minutes on the dark and then with lead citrate for 2 minutes.

Reviewer 2 Report

The review by Giovanna Riparbelli et al., entitled “Cell-to cell interactions during early Drosophila oogenesis: an ultrastructural analysis” addresses the issue of the cell interaction during the drosophila oogenesis using electron microscopy. In drosophila, body axis formation take place early during the development of the egg chamber and relies on the interactive communication between the oocyte and the adjacent follicle cells. Previous studies have shown the importance of the adherens junction (E-cadherin) to established the oocyte location at the posterior pole of the egg chamber. Here the authors investigate this interaction using electron microscopy and described specific structures/ membranes organisation that may be relevant for this process.  

General comments:

The work presented here is very well documented with beautiful electron microscopy images requiring a high level of expertise. Nevertheless, it is very descriptive and could be improved in my view in two ways. Firstly, as the authors have described specific structures of certain regions of the egg chamber (posterior or anterodorsal for later stages), I believe it is very important to compare these regions more deeply with the part of the tissue where the specific structures are not present. This is particularly true for Fig2. The second way to significantly improve the manuscript would be to examine the behaviour of these structures in mutant contexts that may affect them such as a shotgun (E-cadherin) mutant.  

Minor comments:

To reach a wide audience, it would be helpful to assist the reader who is not a specialist in Drosophila oogenesis with a drawing or low magnification panel of the egg chamber to explain the stage and/or location of the structures we observe.

In figure 4 panel D, are the authors sure it is a stage 9, as it looked younger to me. 

Author Response

First at all, we would like to thank the Reviewer for the nice words about our pictures.

As the authors have described specific structures of certain regions of the egg chamber (posterior or anterodorsal for later stages), I believe it is very important to compare these regions more deeply with the part of the tissue where the specific structures are not present. This is particularly true for Fig2.

As required, we add several paragraphs to compare the different regions and structures of the egg chambers.

The second way to significantly improve the manuscript would be to examine the behaviour of these structures in mutant contexts that may affect them such as a shotgun (E-cadherin) mutant. 

We agree with the suggestion of the Reviewer. Analysis of E-cadherin mutants can greatly help our understanding of the membrane organization and dynamics during oogenesis. However, a such EM study requires much time and we, indeed, planned to take this analysis in the next future.

To reach a wide audience, it would be helpful to assist the reader who is not a specialist in Drosophila oogenesis with a drawing or low magnification panel of the egg chamber to explain the stage and/or location of the structures we observe.

As suggested, we added some pictures as insets to the panels Fig.2A, Fig.2C, Fig.2F,  Fig.3A and 3C representing low magnification micrographs of the described egg chambers at different developmental stages.

In figure 4 panel D, are the authors sure it is a stage 9, as it looked younger to me.

It is possible that the oocyte in Fig 4D is younger than we retained in the original description. Therefore, according to the suggestion of the Reviewer we change both the stages in Fig.4 from stage 8 to 7 and from stage 9 to 8.